# Whole Genome Sequencing of the Novel Probiotic Strain *Lactiplantibacillus plantarum* FCa3L

**DOI:** 10.3390/microorganisms11051234

**Published:** 2023-05-07

**Authors:** Olga Karaseva, Georgii Ozhegov, Dilyara Khusnutdinova, Maria Siniagina, Elizaveta Anisimova, Farida Akhatova, Rawil Fakhrullin, Dina Yarullina

**Affiliations:** Institute of Fundamental Medicine and Biology, Kazan Federal University, Kremlevskaya Str. 18, 420008 Kazan, Republic of Tatarstan, Russia; olkakarp66@gmail.com (O.K.); georgii_provisor@mail.ru (G.O.); dilyahusn@gmail.com (D.K.); marias25@mail.ru (M.S.); elizaveta-real@mail.ru (E.A.); akhatovaf@gmail.com (F.A.)

**Keywords:** *Lactiplantibacillus plantarum*, draft genome sequence, genome annotation, probiotic properties

## Abstract

*Lactiplantibacillus plantarum* is best known for its significant adaptive potential and ability to colonize different ecological niches. Different strains of *L. plantarum* are widely used as probiotics. To characterize the probiotic potential of the novel *L. plantarum* FCa3L strain isolated from fermented cabbage, we sequenced its whole genome using the Illumina MiSeq platform. This bacterial isolate had a circular chromosome of 3,365,929 bp with 44.3% GC content and a cyclic phage phiX174 of 5386 bp with 44.7% GC content. The results of in vitro studies showed that FCa3L was comparable with the reference probiotic strain *L. plantarum* 8PA3 in terms of acid and bile tolerance, adhesiveness, H_2_O_2_ production, and acidification rate. The strain 8PA3 possessed higher antioxidant activity, while FCa3L demonstrated superior antibacterial properties. The antibiotic resistance of FCa3L was more relevant to the probiotic strain than that of 8PA3, although a number of silent antibiotic resistance genes were identified in its genome. Genomic evidence to support adhesive and antibacterial properties, biosynthesis of bioactive metabolites, and safety of FCa3L was also presented. Thus, this study confirmed the safety and probiotic properties of *L. plantarum* FCa3L via complete genome and phenotype analysis, suggesting its potential as a probiotic, although further in vivo investigations are still necessary.

## 1. Introduction

*Lactiplantibacillus plantarum* (formerly known as *Lactobacillus plantarum*) is an exceptionally versatile species of lactic acid bacteria (LAB) widely used in food and probiotics production. It dwells in various ecological niches including the human gastrointestinal (GI) tract and different fermented foods such as silage, sauerkraut, kimchi, sourdough, olives, pickles, etc. [1]. The safety of this species is confirmed by its generally recognized as safe (GRAS) status from the United States Food and Drug Administration (US FDA) [2] and Qualified Presumption of Safety (QPS) status from the European Food Safety Authorities (EFSA) [3]. *L. plantarum* is valued in the food industry for its ability to alter favorably the texture, flavor, nutritional quality, and the shelf life of foods [4]. The range of its metabolites with antimicrobial properties includes organic acids (mainly lactic and acetic acid), hydrogen peroxide, bacteriocins (plantaricins), and some others [5,6,7]. The antimicrobial activity of this LAB along with its resistance to the harsh conditions of the human GI tract, adhesive properties, and antioxidant activity makes *L. plantarum* strains an effective probiotic with promising health-promoting effects [8,9,10]. Probiotics are defined as living organisms which, when administered in adequate amounts, confer a health benefit to the host [11]. The incomplete list of disorders which benefit from the application of specific probiotic strains of *L. plantarum* comprises irritable bowel syndrome [12], chronic diarrhea [13], atopic dermatitis [14], vaginal infections [15], cardiovascular disease [16], cancers [17,18], and neurodegenerative diseases [19,20]. Health-promoting effects of *L. plantarum* are based on its promising biological activities, such as ability to alter intestinal flora and inhibit the growth of potential pathogens, modulation of host immune responses, antioxidant, antimutagenic, and antihypertensive activities [8,16].

Currently (April 2023), 887 genome assembly and annotation reports of *L. plantarum* strains are cataloged in the NCBI database, with *L. plantarum* WCFS1 being the first sequenced genome of *Lactobacillus* species in 2003 and now being the most studied one [21,22]. To date, after hundreds of bacterial genomes were unraveled, routine traditional microbiological methods are being replaced by ‘omics’ approaches (also referred to as ‘probiogenomics’) for more comprehensive and rapid understanding of probiotic and functional properties of *L. plantarum*, especially its specific interactions with the host [10,23]. High genome variability of the former *Lactobacillus* genus was one of the rationales for its reclassification into 25 genera [1]. Unlike most LAB, the genome of *L. plantarum* is not reduced and carries many genes which have been acquired by horizontal gene transfer via mobile elements, such as prophages, transposons, integrons, and plasmids [24]. Surprisingly, among all *Lactobacillus* species strains of *L. plantarum* own the utmost diverse functional genomes that facilitate metabolic flexibility and high adaptive potential and allow them to colonize a variety of environments [24,25,26]. The probiotic properties of *L. plantarum* are also strain-specific [24,27].

In this study, genome sequencing of *L. plantarum* FCA3L isolated from fermented cabbage was performed using the Illumina MiSeq platform for species classification, gene prediction, and functional annotation, giving new insights into the probiotic potential of the strain. In addition, we experimentally evaluated some probiotic properties of *L. plantarum* FCA3L such as acid and bile tolerance, adhesiveness, antibacterial and antioxidant activities, hydrogen peroxide production, acidification rate, and resistance to antibiotics.

## 2. Materials and Methods

### 2.1. The Lactiplantibacillus Strains

*L. plantarum* strain FCa3L was previously isolated from the sauerkraut collected from a local market (Kazan, Tatarstan, Russia) in September 2016. It was assigned to *Lactobacillus plantarum* by MALDI-TOF mass spectrometry (MALDI Biotyper system, Bruker Daltonik, Germany) [28]. *L. plantarum* 8PA3, approved as a probiotic strain (Biomed, Russia), was used as a reference [29].

### 2.2. Species Identification

For most accurate species identification, the 16S rRNA gene was amplified by PCR method using universal 16S rRNA bacterial primers 27F and 1392R, as described earlier [30]. The 1.4 kb DNA fragments were purified from the agarose gel after electrophoresis and sequenced on an ABI Prism 3730 sequencer (Applied Biosystems, Waltham, MA, USA). The species was identified on the basis of 16S rRNA gene sequence’s similarity obtained using NCBI database and BLAST algorithm (https://www.ncbi.nlm.nih.gov/BLAST accessed on 19 April 2023).

### 2.3. Genomic DNA Extraction and Sequencing

For whole-genome sequencing, *L. plantarum* FCa3L was cultured in 30 mL de Man, Rogosa, Sharpe (MRS) broth (HiMedia, India) under microaerophilic conditions at 37 °C for 24 h, harvested by centrifugation at 5000 rpm for 20 min and washed with sterile PBS. Genomic DNA extraction and purification were carried out using the ZymoBIOMICS DNA Miniprep Kit (Zymo Research, Irvine, CA, USA). The sediment was suspended in 0.01 M Tris-HCl (pH 8.0) with lysozyme (20 mg/mL) and incubated at 37 °C for 1 h. Further extraction was performed according to the manufacturer’s protocol. Genomic DNA integrity was assessed by electrophoresis in 0.8% agarose gel and using OD260/OD280 ratio on Nanodrop 2000. For Illumina MiSeq sequencing, total DNA was sheared to fragments ranging between 300 and 500 bp with an average size around 400  bp using the Covaris S220. Then, DNA libraries were prepared using NEBNext Ultra II Kit for Illumina (New England BioLabs, Ipswich, MA, USA) according to the manufacturer’s recommendations. The quality of the final DNA library was evaluated on the Agilent 2100 Bioanalyzer (Agilent Technologies, Santa Clara, CA, USA). Sequencing was performed on Illumina MiSeq platform (300 bp paired-end mode). The quality of raw sequence reads was evaluated by FastQC package (v0.11.9) [31]. The raw data reported in this article are available in the NCBI Sequence Read Archive at https://www.ncbi.nlm.nih.gov/bioproject, accessed on 19 April 2023. The BioProject accession number is PRJNA786732.

### 2.4. Genome Assembly, Annotation, Phylogenetic and Functional Analysis

The Unicycler genome assembler v. 0.4.8 and SequenceScanner v. 1.0 (Applied Biosystems) were used to assemble sequencing data generated by Illumina MiSeq [32]. The annotation of the *L. plantarum* genome was performed via EggNOG v. 4.5 (http://eggnog5.embl.de/#/app/seqscan, accessed on 19 April 2023), RAST (https://rast.nmpdr.org/, accessed on 19 April 2023), and NCBI Prokaryotic Genome Annotation Pipeline (PGAP) (https://www.ncbi.nlm.nih.gov/genome/annotation_prok/, accessed on 19 April 2023) for prediction of protein-coding genes [33]. Geneious Prime 2023.1 and its plugins were used for bioinformatic analysis [34]. A whole genome-based phylogenetic tree was constructed based on publicly available *Lactobacillus* strains, including *L. plantarum* Heal 19, CNEI-KCA5, AMT74419, LS/07, X7022, 202195, CACC 558, SRCM101511, SRCM102737, SRCM101518, SRCM101222, SRCM101187, SRCM101105, SRCM100995, SRCM100442, SRCM100440, SRCM100438, 8P-A3, 83-18, KCCP11226, LLY-606, pc-26, SKO-001, JCM8341, EM, IRG1, SCB0151, 022AE, ST, GR1186, W2, MSD1, DW12, CHE37, GR1184, GR1187, XJ25, ATCC 202195, NCIMB8826, LRCC5314, KLDS1.0386, BK-021, AR195, Lp900, KM2, SHY 21-2, MK55, PMO08, DSM 20174, SK156 (https://www.ebi.ac.uk/Tools/phylogeny/simple_phylogeny/, accessed on 19 April 2023).

The protein-coding genes were blasted against the databases of Swiss-Prot (http://www.uniprot.org/, accessed on 19 April 2023), GO (http://www.geneontology.org/, accessed on 19 April 2023), eggNOG/COG (http://www.ncbi.nlm.nih.gov/COG/, accessed on 19 April 2023), KEGG (http://www.genome.jp/kegg/, accessed on 19 April 2023), and RAST (https://rast.nmpdr.org/, accessed on 19 April 2023) to perform gene function analysis.

### 2.5. Atomic Force Microscopy

For atomic force microscopy (AFM), bacteria were grown in 2 mL MRS broth on 34-mm plates (TC-treated, Eppendorf) at 37 °C for 24 h, washed with milliQ water and fixed with 0.1% aqueous solution of glutaraldehyde for 24 h. After subsequent washing with milliQ water, the plates were air-dried and imaged using a Dimension Icon microscope (Bruker, Billerica, MA, USA) operating in the PeakForce Tapping mode with a probe (ScanAsyst-Air, nominal length 115 μm, nominal tip radius 2 nm, spring constant 0.4 N/m, Bruker). The images were obtained at 512 lines/scan at 0.8–0.9 Hz scan rate. The images were acquired in height (topography), peak force error, and adhesion channels. The AFM data were processed and analyzed using Nanoscope Analysis v.1.7 software (Bruker).

### 2.6. Assessment of Probiotic Properties of L. Plantarum FCa3L

#### 2.6.1. Acid and Bile Tolerance

Overnight cultures were harvested by centrifugation and washed twice with physiological saline. Cells were resuspended to a final cell concentration of approximately 10^8^ CFU/mL in physiological saline containing 2% ox gall (ZAO NICF, St. Petersburg, Russia) or acidified to pH 2 using 1 N HCl and in physiological saline without additives for control. After the incubation for 2 or 4 h at 37 °C with shaking (180 rpm) to simulate intestinal conditions, acid and bile tolerances were evaluated by plate count method. MRS agar plates were incubated at 37 °C for 48 h.

#### 2.6.2. Antagonistic Activity

Antagonistic activity towards bacterial pathogens and common bacteria present in the GI tract was examined by agar block test described in [35]. Briefly, overnight LAB cultures in MRS broth were distributed onto MRS agar and incubated anaerobically (Anaerogas gaspack, NIKI MLT, Russia) at 37 °C for 48 h. Agar blocks containing the isolate growth (~0.5 cm^3^) were aseptically cut and transferred to Lysogeny agar (LB agar) (1% tryptone, 0.5% yeast extract, 0.5% NaCl, pH 8.5) inoculated with ~10^7^ cells of test bacteria (indicator bacteria): gram-negative *Escherichia coli* MG1655 (K-12), *Pseudomonas aeruginosa* ATCC 27853, *Klebsiella pneumonia*, *Serratia marcescens*, *Morganella morganii* MM190 (clinical isolates), and gram-positive *Staphylococcus aureus* ssp. *aureus* ATCC 29213, *Bacillus cereus*, *Micrococcus luteus*, and *Enterococcus faecalis* (clinical isolates). All clinical isolates used in this study were obtained from Kazan Institute of Epidemiology and Microbiology (Kazan, Russia) except for *M. morganii* MM190 which was kindly provided by Dr. Ayslu Mardanova [36] and *K. pneumoniae* and *S. marcescens* provided by Institute of Medical Microbiology (Giessen, Germany). After 24 h of incubation at 37 °C, zones of bacterial growth inhibition were measured from the edge of the agar block to the edge of the inhibition zone. The inhibitory effect of agar blocks with MRS agar was used as negative control. Each test was performed in triplicate.

#### 2.6.3. Acidification Rate

The acidification rate of LAB strains was evaluated by measuring Total Titratable Acidity (TTA) of the cell-free supernatant, obtained by centrifugation of 48 h LAB culture in MRS broth. The cell-free supernatants were diluted twofold with distilled water and titrated with 0.1 N NaOH using phenolphthalein as an indicator. TTA (mmol acid/mL) was calculated by multiplication of NaOH needed for titration by 0.09 [37].

#### 2.6.4. H_2_O_2_ Determination

The H_2_O_2_ production was studied by culturing microorganisms in TMB-MRS agar plates. The plates were prepared with MRS agar supplemented with 3,3′,5,5′-Tetramethyl-Benzidine (TMB) (Sigma-Aldrich, St. Louis, MO, USA) to a final concentration of 0.25 mg/mL and horseradish peroxidase to a final concentration of 0.01 mg/mL. After incubation at 37 °C in a 5% CO_2_ atmosphere for 48 h, the plates were exposed to the air. Colonies able to produce H_2_O_2_ develop a color of blue or brown. According to the color intensity, the strains were classified as strong (blue), medium (brown), weak (light brown), or negative (white colonies) producers [38].

For quantitative H_2_O_2_ determination, overnight LAB cultures were harvested by centrifugation, cells were washed twice with PBS and resuspended in 500 µL Lysogeny broth (LB). After 4 h incubation with shaking at 200 rpm at 37 °C, the suspension was centrifuged 3 min at 7000 rpm and H_2_O_2_ concentration was measured in the resulting supernatant by the method based on oxidation of ferrous to ferric ion in the presence of xylenol orange (XO). PeroxiDetect^TM^ Kit (Sigma-Aldrich) was used following the manufacturer’s instructions.

#### 2.6.5. MATS Method

Physicochemical properties of bacterial cell surface were evaluated using Microbial Adhesion To Solvents (MATS) method [39] with some modifications [40]. Chloroform, n-hexadecane, and ethyl acetate (all Sigma-Aldrich) were used to assess the electron donor (basic), hydrophobic, and electron acceptor (acidic) characteristics of bacterial surface, respectively.

#### 2.6.6. Autoaggregation Assay

Autoaggregation assay was performed as described in [41]. Briefly, overnight cultures were harvested by centrifugation and washed twice with PBS. Cells were resuspended in PBS to an optical density of 0.5 at 600 nm (A_0_) (approximately 10^7^–10^8^ CFU/mL). Bacterial cell suspensions (4 mL) were incubated at room temperature in tubes for 4 h or 24 h without shaking. The aqueous phase was gently taken out to measure its absorbance at 600 nm (A_1_). Autoaggregation percentage was calculated as (1 − A_1_/A_0_) × 100.

#### 2.6.7. Antibiotic Resistance

Antibiotic resistance was assessed by the disk diffusion method, as described earlier [28]. Antibiotic discs were purchased from Scientific Research Centre of Pharmacotherapy (St. Petersburg, Russia). Strains were classified either as resistant (R), moderately susceptible (MS), or susceptible (S) based on zones of growth inhibition according to [42,43].

#### 2.6.8. Antioxidant Activity

Antioxidant activity of the LAB strains was screened by measuring their DPPH (Sigma-Aldrich) free radical scavenging activity according to the method of [44], with some modifications. The LAB strains were cultured in MRS broth at 37 °C for 24 h and then harvested by centrifugation. The resulting supernatant was mixed 1:1 (*v/v*) with ethanol DPPH solution (0.1 mM) and incubated for 30 min at 25 °C in the dark. The optical absorbance (A) at 517 nm was measured using microplate reader xMarkTM (BioRad, USA). The DPPH scavenging activity was defined as scavenging activity (%) = [1 − (A_sample_)/A_control_] × 100%, where A_sample_ is the A at 517 nm of the sample after incubation, and A_control_ is the absorbance of pure ethanol DPPH solution (0.1 mM).

### 2.7. Statistical Analysis

All experiments were independently conducted two or three times, and each assay was performed in triplicate. Quantitative data are presented as the means ± standard deviations that were analyzed using GraphPad Prism 5. Statistical differences between mean values were determined using Student’s t test at a significance level of *p* < 0.05.

## 3. Results and Discussion

### 3.1. L. plantarum FCa3L Identification and Morphology

Identification of the FCa3L strain based on 16S rRNA analysis revealed its assignment to *L. plantarum* and thus corresponded to species commonly associated with sauerkraut [1]. In AFM, cells appeared as short straight rods with a size 0.8−1.0 μm × 2−3 μm occurring singly and in short chains. The cells lacked glycocalyx, flagella, or fimbriae and thus exhibited typical morphology of *L. plantarum* (Figure 1). Colonies on MRS agar were small (approximately 1–2 mm in diameter), round, convex, smooth, with entire margins, and white.

### 3.2. The Genome of L. plantarum FCa3L

In this article, we present a description of the genome sequencing data and the draft genome of *L. plantarum* FCa3L. Genome sequencing was performed using the Illumina MiSeq platform and a total of 40 Mb raw data was obtained. The genome sequence of *L. plantarum* FCa3L at 30× coverage was submitted to the NCBI database (NZ_JAJSYR010000001.1). The genome size was 3,365,929 bp, comprising 55 contigs with GC content of 44.3%. The quality parameters for the genome assembly such as N50 value (184003) and L50 value (8) indicated the average quality of the assembled genome. The *L. plantarum* FCa3L genome contained 3120 coding sequences (CDS), 3193 genes, 3071 proteins, 49 pseudogenes, 3 rRNAs, 66 tRNAs, and 4 other RNAs. The genome size and GC content of *L. plantarum* FCa3L corresponded to that reported in *L. plantarum* strains. The genome size within *L. plantarum* species ranges from 2.91 to 3.7 Mb, being one of the largest genomes within the lactobacilli group [9]. According to the NCBI database, the median total genome length of *L. plantarum* is 3.27272 Mb with a median protein count of 2997 and GC content of 44.5% (https://www.ncbi.nlm.nih.gov/genome/?term=txid1590[orgn], accessed on 30 April 2023). Large genomes of *L. plantarum* are believed to be directly related to its ecological flexibility and remarkable adaptability to diverse habitats [24]. The circular complete genome draft shown in Figure 2 was constructed using Geneious Prime. To validate this assembly, we mapped the obtained scaffold onto the complete genome of the reference strain *L. plantarum* 8PA3 (ASM440304v2), resulting in a mapping rate up to 100% (Figure 3). The whole genome-based phylogenetic tree showed that *L. plantarum* strains DSM 20,174 and SK156 are the closest evolutionary relatives of *L. plantarum* FCa3L (Appendix A). These *L. plantarum* strains were isolated from pickled cabbage and traditional Korean food, respectively [45], and therefore share a similar origin with *L. plantarum* FCa3L. The RAST server-based annotation of the FCa3L genome resulted in a total of 232 subsystems with 24% subsystem coverage (Figure 4). The subsystem category distribution of the genes assigned to different subsystems indicated the highest genes assigned to the metabolism of carbohydrates (232 genes), followed by the metabolism of amino acids and derivatives (173 genes), protein metabolism (124 genes), cofactors, vitamins, prosthetic group, and pigments (103 genes). However, function unknown was the most common term (2526 genes which comprise 76% of the genome).

In search for bioactive metabolites which can contribute to the probiotic potential of *L. plantarum* FCa3L in its genome sequence we identified genes, encoding biosynthesis of B-group vitamins: biotin (*bioA*, *bioD*, *bioB*, *bioW*, *bioC*, *bioH*, *bioG*, *bioK*, *bioZ*, *bioN*, *bioM*, *bioX*, *bioR*), thiamine (*thiF*, *thiI*, *thiH*, *thiO*, *thiG*, *thiC*, *thiD*, *thi5*, *thiE*, *thiM*), riboflavin (*ribK*, *RK, RSK*, *RSAa*), and folic acid (*folP*, *folA3*, *folB*, *folK*, *folE1*, *tilS*). The use of vitamin-producing starters or probiotics represents a promising strategy to fortify fermented foods and increase the health-promoting effect of probiotics [46]. However, future in silico and in vitro studies should elucidate how the genetic capability to biosynthesize vitamins correlates with the distinct biochemical biosynthesis pathways. We also analyzed genetic data related to safety assessment such as virulence factors and toxin-encoding genes but did not reveal any risk-related sequences in the genome of *L. plantarum* FCa3L. A cyclic 5386 base contig with GC content of 44.7% was identified as *Escherichia* phage phiX174 [47]. Evanovich et al. examined mobile elements in the complete genomes of *L. plantarum* available in the GenBank sequence database and showed that the most encountered bacteriophages were Sha1 and Phig while phiX174 was not detected at all [24]. It is known that bacteriophages infecting probiotic *Lactobacillus* strains do not pose a threat to a consumer’s health, but quite on the contrary are considered a beneficial component of probiotics that target the pathogenic bacteria and support the natural human microbiota [48].

### 3.3. Probiotic Properties of L. plantarum FCa3L

#### 3.3.1. Tolerance of *L. plantarum* FCa3L to Acid and Bile In Vitro

An important prerequisite to consider a strain as a probiotic is its ability to survive the harsh conditions of the human GI tract. A probiotic strain must initially withstand the acidic conditions of the stomach, and then tolerate exposure to bile acids and salts in the small intestine [10]. *L. plantarum* FCa3L was able to tolerate bile salts at 2% ox gall for 2 h and significantly reduced its viability to 65% after 4 h exposure to 2% ox gall. After submitting the *L. plantarum* FCa3L to HCl solution with pH 2, we observed a reduction of the survival rate to 12.8% which continued to slightly decrease during next 2 h. Overall, the tolerance of *L. plantarum* FCa3L to the GI passage corresponded to that of the reference strain *L. plantarum* 8PA3 (Table 1).

#### 3.3.2. The Antibacterial, Acidifying, and Antioxidant Activities of *L. plantarum* FCa3L

The antibacterial activity of *L. plantarum* FCa3L was examined using nine bacterial indicator strains, both gram-negative and gram-positive (Table 2). The antagonistic activity of *L. plantarum* FCa3L preceded that of the reference strain *L. plantarum* 8PA3 towards *E. coli*, *P. aeruginosa*, *M. morganii*, *S. aureus*, and *B. cereus*. These five strains were the most sensitive to the antimicrobial activity of *L. plantarum* FCa3L of all tested indicator strains. The exceptionally strong antagonism of *L. plantarum* 8PA3 was manifoldly proved during its long history (over 40 years) of the clinical implication in several pharmaceutical probiotics distributed in Russia, Ukraine, Belarus, and some other countries [29]. *L. plantarum* 8PA3 is known to prevent the growth of *Streptococcus pyogenes, Streptococcus agalactiae, Staphylococcus aureus*, *E. coli*, *P. aeruginosa*, *Klebsiella oxytoca*, and *K. pneumoniae* [29,49]. According to our results, *L. plantarum* FCa3L had greater antagonistic activity against some pathogens than *L. plantarum* 8PA3, suggesting its promising probiotic potential.

To determine the origin of the detected antimicrobial activity, the ability to produce hydrogen peroxide and TTA were tested. Both LAB strains demonstrated comparable levels of TTA and similar ability to produce H_2_O_2_, as was detected in TMB and XO assays (Table 1). *L. plantarum* is a facultative heterofermentative LAB and therefore is expected to produce primarily lactic acid when grown on MRS broth with glucose. We demonstrated that *L. plantarum* FCa3L was able to produce H_2_O_2_ and decrease pH via the production of organic acids (primarily lactic acid) to the same extent as the reference strain. Considering its higher antagonistic activity compared to *L. plantarum* 8PA3, we suggest that the antagonistic activity of *L. plantarum* FCa3L along with acidification and H_2_O_2_ production is determined by other factors, in particular, production of bacteriocins. In the genome of *L. plantarum* FCa3L, at least one plantaricin gene was identified (CDS 214437..214736), encoding a bacteriocin immunity protein (lp_2952 in the genome of *L. plantarum* WCFS1). Further experimental analysis and bioinformatics of bacteriocins production in *L. plantarum* FCa3L can contribute to the unraveling of the nature of the strong antagonistic activity of *L. plantarum* which is still incompletely understood.

Along with the ability to produce H_2_O_2_, *L. plantarum* FCa3L exhibited an antioxidant activity since its cell-free supernatant was able to scavenge DPPH free radicals. However, its DPPH-radical scavenging activity was two times lower compared to the reference strain *L. plantarum* 8PA3 (Table 1). We may speculate that *L. plantarum* protect themselves against oxygen radical injury through radical scavenging activity.

#### 3.3.3. Cell Surface Properties and Adhesiveness

Bacterial cell surface hydrophobicity is one of the most important factors that influence bacterial adhesion [50]. The hydrophobic cells adhere strongly to hydrophobic surfaces such as mucus [51]. The surface hydrophobicity of lactobacilli cells assessed by measuring microbial adhesion to hexadecane corresponded to moderate hydrophilicity (10–29%), according to the ranking offered in [52]. It is presumed that the hydrophilic surface is a consequence of cell-wall associated carbohydrates [41,53].

According to the increased affinities with chloroform and decreased with ethyl acetate, both strains demonstrated basic and electron donor character of the bacterial surface, which is probably related with the presence of a carboxylic (-COO^−^) and hydrogen sulfite (-HSO_3_^−^) groups [53].

Therefore, the studied strain is characterized by basic and electron donor character as confirmed by its hydrophilic cell surface properties. A similar observation was previously obtained by [53] for eight *Lactobacillus* strains. The initial adhesion interaction is believed to involve nonspecific mechanisms and therefore is dependent on physical characteristics of the bacterial cell surface such as hydrophobicity and electron donor–acceptor properties [54]. Although two strains in this study had similar cell surface characteristics, nonspecific adhesion of *L. plantarum* 8PA3 was significantly higher as compared with *L. plantarum* FCa3L. Perhaps, this difference originated from different surface roughness as was determined by AFM. Increased surface roughness might have a negative effect on adhesive ability, as it hinders the cell-to-cell contact, and as a result, *L. plantarum* 8PA3 with a smoother surface demonstrated stronger nonspecific adhesion. Additionally, bacteria can modify their cell surfaces regarding hydrophobicity and surface roughness in response to changes in environmental conditions and growth phases [55,56].

Autoaggregation (ability to form floccules) is a key factor for colonization of mucosa in the gastrointestinal and urogenital tracts [41]. Bacteria with the ability to autoaggregate remain in the intestines for a longer time and thus better exert their probiotic effects [57]. *L. plantarum* FCa3L had medium autoaggregation capacity (33% < Autoagg. < 66%), while *L. plantarum* 8PA3 had low autoaggregation capacity (Autoagg. < 33%) after 4 h incubation. The autoaggregation capacity of lactobacilli usually ranges from low to moderate [58,59]. The standard probiotic cultures such as *L. johnsonii* LA1, *L. acidophilus* LA7, and *L. rhamnosus* GG after 5 h of incubation self-aggregated at the level of 40.4 ± 0.4%, 46.5 ± 2.0%, and 41.39 ± 3.30%, respectively [58,60]. After 24 h of incubation, both strains demonstrated high levels of autoaggregation (Autoagg. > 66%). High nonspecific adhesion measured by AFM might result in flocculation of the cells and high autoaggregation values, but data obtained in this work are not consistent with this pattern (Table 1).

We queried the *L. plantarum* FCa3L genome sequence for genes whose products function in bacterial adhesion. Expected functions were confirmed in the NCBI database with BLASTp. Using GO and eggNOG/COG databases, the candidate gene for sortase (CDS 161440..162144) was revealed when gene *srt*A from *L. plantarum* WCFS1 was used as a query [61]. Similarly, candidate genes for fibronectin-binding protein (CDS 76690..77562) and glucosyltransferase (CDS 53439..54938) were identified when *fbp*A from *L. acidophilus* NCFM (Lba1148) and *gtf*A from *L. reuteri* TMW1.106 were used as a query, respectively [62].

#### 3.3.4. Antibiotic Susceptibility

The relevance of antibiotic susceptibility of probiotic strains is determined first by potential spread of resistance genes from probiotics to pathogens and then by the possibility of combining resistant probiotic bacteria with antibiotic treatment [63]. *L. plantarum* FCa3L was susceptible to ampicillin, chloramphenicol, clindamycin, erythromycin, rifampicin, and tetracycline and exhibited resistance to vancomycin, ciprofloxacin, and aminoglycosides (amikacin, kanamycin, streptomycin, and gentamicin) (Table 3). The resistance to aminoglycosides, vancomycin, and ciprofloxacin is usually intrinsic in lactobacilli and thus encoding chromosomal genes are considered not to be transferable to other bacteria [63]. Surprisingly, the reference strain *L. plantarum* 8PA3 revealed an antibiotic resistance pattern not typical for lactobacilli with susceptibility to vancomycin and moderate susceptibility to the inhibitors of protein synthesis clindamycin and erythromycin. The latter is the most concerning because resistance to erythromycin is often acquired in lactobacilli and therefore potentially transferrable to new hosts [64].

Then, we checked the presence of potential antibiotic resistance genes in the genome of *L. plantarum* FCa3L and revealed genes of tetracycline resistance (*tetK*, *tetM*, *tetO*, *tetQ*, *tetW*, *tetlike*, *tetlike2*), oxytetracycline resistance (*otrA*), fluroquinolone resistance (*parC*, *parE*, *gyrA*, *gyrB*), and genes encoding multiple drug resistance (*cmeA*, *cmeB*, *cmeC*, *macA*, *macB*, *mtrF*, *acrB*, *mexD*, *mexC*).

## 4. Conclusions

In this work, a genomic analysis combined with experimental studies on a novel potential probiotic strain *L. plantarum* FCa3L allowed us to obtain its more comprehensive probiotic profile which included an accurate taxonomic assignment, safety assessment, and probiotic traits, such as the ability to survive the human GI tract, adhesion, antibacterial and antioxidant activity, production of bioactive metabolites, and antibiotic resistance. The whole genome sequence of *L. plantarum* FCa3L opens perspectives for new studies to be carried out to identify genetic factors and mechanisms related to its beneficial effects as a probiotic. Considering genomic data as an a priori prerequisite for successful probiotic implementation, results of an investigation a posteriori of the primary probiotic properties of *L. plantarum* FCa3L revealed its relatively high acid and bile tolerance, adhesiveness, hydrogen peroxide production, and acidification rate. The reference probiotic strain *L. plantarum* 8PA3 possessed more pronounced antioxidant properties, while *L. plantarum* FCa3L demonstrated superior antibacterial activity. The antibiotic resistance profile of FCa3L was more relevant to the probiotic strain than that of 8PA3, although a number of silent antibiotic resistance genes were identified in the genome of the novel isolate. Overall, the results of this study nominate *L. plantarum* FCa3L for the role of a promising probiotic candidate with high commercial and biotechnological relevance, but first, its health-promoting benefits and safety should be investigated in vivo in an animal model.

## Figures and Tables

**Figure 1 microorganisms-11-01234-f001:**
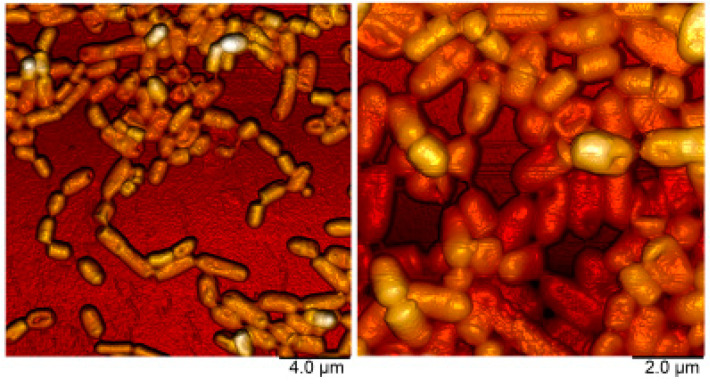
AFM topography images demonstrating the morphology of *L. plantarum* FCa3L.

**Figure 2 microorganisms-11-01234-f002:**
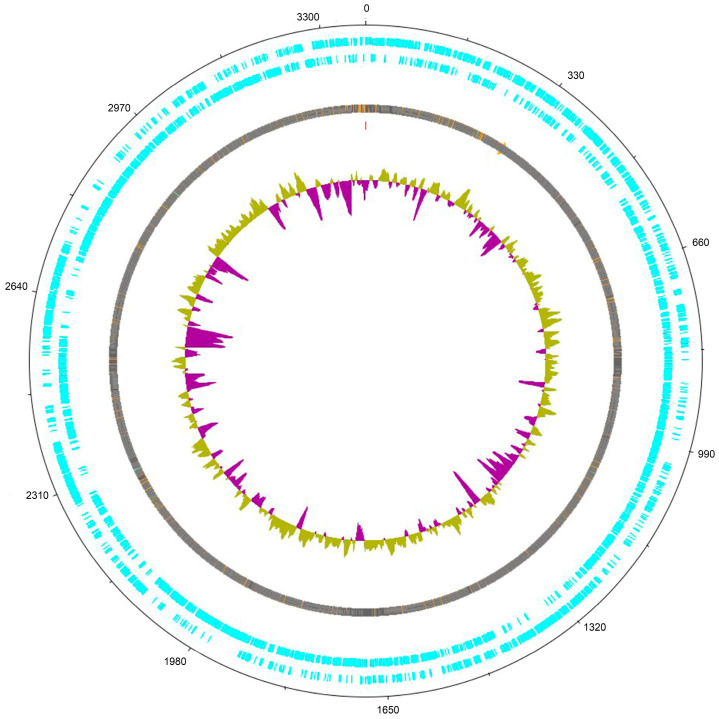
Circular genome plot of *L. plantarum* FCa3L generated using Geneious Prime. Blue tracks show the coding genes in the positive and negative strands, respectively. Purple-yellow track is the GC plot: purple peaks are a segment of genomes with GC% below average, and yellow peaks are segments with GC% above average.

**Figure 3 microorganisms-11-01234-f003:**
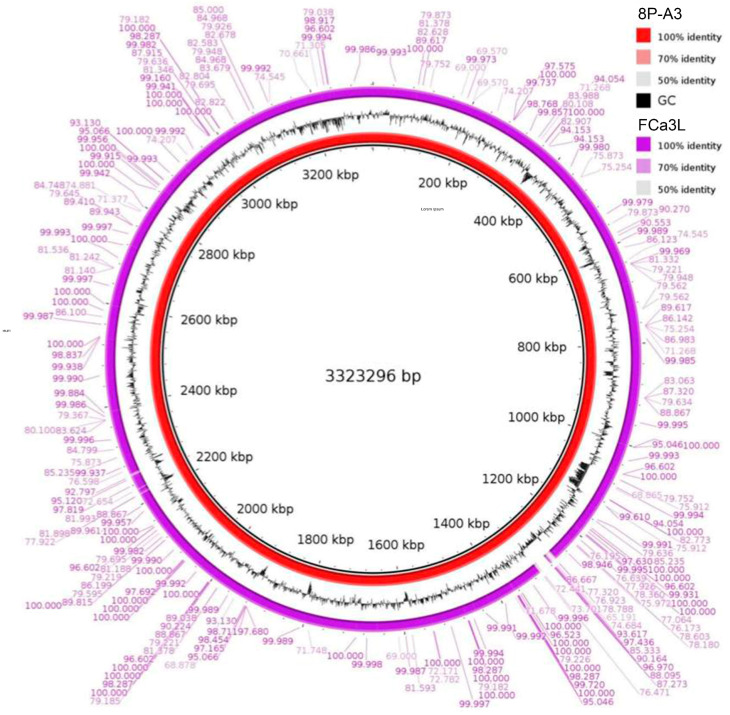
Mapping of the *L. plantarum* FCa3L scaffold onto a reference complete genome of *L. plantarum* 8PA3 using Geneious Prime.

**Figure 4 microorganisms-11-01234-f004:**
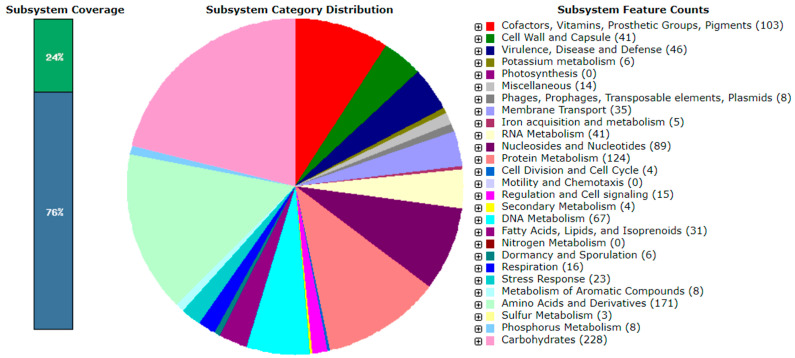
An overview of the subsystem categories assigned to the genes predicted in the genome of *L. plantarum* FCa3L by the RAST server.

**Table 1 microorganisms-11-01234-t001:** Probiotic properties of *L. plantarum* strains.

Probiotic Properties	FCa3L	8PA3
Survival in GI tract, % ^1^		
Ox gall 2%		
2 h	103.8	113.7
4 h	65.0	52.9
HCl, pH = 2		
2 h	12.8	15.2
4 h	11.3	10.0
Adhesion		
% of adhesion ± SD to:		
Hexadecane	21.9 ± 1.6	24.9 ± 8.9
Ethyl acetate	25.9 ± 3.7	22.9 ± 3.2
Chloroform	92.8 ± 3.2	87.9 ± 2.9
AFM analysis of the surface structure and nonspecific adhesion of bacterial cells		
Surface roughness (500 × 500 nm), Sq, nm	4.5 ± 1.6 *	1.9 ± 0.6
Surface roughness (500 × 500 nm), Sa, nm	3.4 ± 1.3	1.5 ± 0.5
Nonspecific adhesion of the biofilm surface (20 × 20 µm), nN	3.1 ± 1.1	6.8 ± 2.7
Nonspecific adhesion of the cell surface (500 × 500 nm), nN	2.0 ± 0.1 *	11.4 ± 2.8
Auto-aggregation, %		
4 h	58.7 ± 7.9 *	23.1 ± 4.2
24 h	67.2 ± 3.7	72.7 ± 8.4
Total titratable acidity (TTA), mmol/mL	1.46 ± 0.15	1.37 ± 0.23
H_2_O_2_ production		
TMB assay	Present	Present
XO assay, µM	53.3 ± 2.3	58.0 ± 5.0
Antioxidant activity, %	9.1 ± 4.7	20.5 ± 5.9

^1^ The experimental data scatter did not exceed 5%. * means statistically significant difference (*p* < 0.05) as compared to the reference strain *L. plantarum* 8PA3.

**Table 2 microorganisms-11-01234-t002:** Antimicrobial activity of *L. plantarum* strains (growth inhibition, mm).

Indicator Microorganisms	FCa3L	8PA3
Gram-positive		
*Bacillus cereus* (Clinical isolate)	8.60 ± 1.14 *	2.75 ± 0.50
*Enterococcus faecalis* (Clinical isolate)	3.40 ± 0.89	1.25 ± 0.96
*Micrococcus luteus* (Clinical isolate)	1.20 ± 0.84	0.75 ± 0.50
*Staphylococcus aureus* ATCC 29213	7.20 ± 0.45 *	1.75 ± 0.95
Gram-negative		
*Escherichia coli* K-12	5.20 ± 0.84 *	3.00 ± 0.82
*Pseudomonas aeruginosa* ATCC 27853	5.00 ± 0.71 *	1.25 ± 0.50
*Klebsiella pneumonia* (Clinical isolate)	2.80 ± 0.45	2.00 ± 1.41
*Morganella morganii* MM190 (Clinical isolate)	8.50 ± 0.00 *	5.25 ± 0.50
*Serratia marcescens* (Clinical isolate)	1.80 ± 0.45	1.50 ± 0.57

* means statistically significant difference with probiotic strain *L. plantarum* 8PA3 (*p* < 0.05).

**Table 3 microorganisms-11-01234-t003:** Antibiotic resistance of *L. plantarum* strains.

Antibiotics	Amount per disc, μg	LAB Strains	
		FCa3L	8PA3
Ampicillin	10	S (25.0 ± 3.5)	S (43.0 ± 2.8)
Amikacin	30	R (5.0 ± 0.0)	R (7.0 ± 2.8)
Chloramphenicol	30	S (28.0 ± 2.8)	S (19.0 ± 3.0)
Ciprofloxacin	5	R (6.0 ± 0.0)	R (7.0 ± 2.0)
Clindamycin	2	S (12.5 ± 3.5)	MS (11.5 ± 2.1)
Erythromycin	15	S (20.5 ± 1.4)	MS (14.0 ± 3.5)
Gentamicin	10	R (5.5 ± 0.7)	R (5.0 ± 0.0)
Kanamycin	30	R (6.0 ± 0.0)	R (6.0 ± 0.0)
Rifampicin	5	S (21.0 ± 0.0)	S (20.5 ± 0.7)
Streptomycin	30	R (5.5 ± 0.7)	R (7.0 ± 2.8)
Tetracycline	30	S (19.0 ± 1.4)	S (26.0 ± 4.0)
Vancomycin	30	R (5.0 ± 0.0)	S (21.0 ± 4.2)

Diameters of inhibition zones, mm (means ± SD of three trials) were interpreted as susceptible (S), moderately susceptible (MS), or resistant (R).

## Data Availability

The data will be made available upon reasonable request.

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
