# Peer review of "Whole Genome Sequencing of the Novel Probiotic Strain Lactiplantibacillus plantarum FCa3L"

_microorganisms, 2023, doi:10.3390/microorganisms11051234_

Round 1
Reviewer 1 Report
Dear Editor.
The manuscript has been written well and constructed with high scientific standards. Its recommended fro publications on this journal
# Define abbreviations upon the first appearance in the text especially names of microbes
# Gram should not be in italics in Table 2.
Minor editing of English language required
Author Response
Dear Reviewer,
Thank you. Writing of Gram in Table 2 was corrected. Names of microbes are written in the manuscript in accordance with the rules of nomenclature of bacteria. Namely, if the genus name has already been stated, it is further abbreviated to the first letter of the generic name. If species are listed belonging to several genera which have the same initial letter, the generic name is used in full (Cowan M. Kelly. Microbiology: a systems approach, 3rd ed., McGraw-Hill. 2012; Lapage SP, Sneath PHA, Lessel EF, et al., editors. International Code of Nomenclature of Bacteria: Bacteriological Code, 1990 Revision. Washington (DC): ASM Press; 1992, https://www.ncbi.nlm.nih.gov/books/NBK8816/). Corresponding corrections were done in lines 76, 313, and 319.
Reviewer 2 Report
Lactiplantibacillus plantarum is best known for its significant adaptive potential and ability
to colonize different ecological niches. Different strains of L. plantarum are widely used as probiotics.
Authors in order to characterize the probiotic potential of the novel L. plantarum FCa3L strain isolated from
fermented cabbage we sequenced its whole genome using Illumina MiSeq platform.
Well-structured and organized manuscript
English is good no mistakes were found
An adequate number of figures and tables were found
Figures are depicted with utmost care and clarity
The literature review is also fine
Conclusions based on data availability complemented each other well.
Overall, the results of this study nominate L. plantarum FCa3L for a role of a promising
probiotic candidate with high commercial and biotechnological relevance, but first its
health-promoting benefits and safety should be investigated in vivo in an animal model.
Accept it as it was, nothing to revise.
Author Response
Dear Reviewer,
Thank you for your appreciation.
Reviewer 3 Report
In the manuscript entitled “Whole genome sequencing of the novel probiotic strain Lactiplantibacillus plantarum FCa3L” the authors investigated the probiotic properties of Lactiplantibacillus plantarum FCa3L by sequencing its whole genome using Illumina MiSeq platform. This is an interesting topic, and it is an area that really needs our attention. Overall, the work is well done, carefully thought out, and performed. However, there are still some areas of the article that need to be revised.
1. Keywords:
-Please write in alphabetic order.
2. Introduction:
-lines 29: write the abbreviation L. plantarum in brackets for the first time, and then always use the abbreviated form. The same in abstract (line 10).
-lines 47-50: The health-promoting effects of L. plantarum are extensively studied in view of innovative valorisation strategies including the bioconversion of L. plantarum plant by-products to obtain enzymes, polysaccharides, nutraceuticals and beverages. It is also important in sourdough fermentation-based strategies for making baked goods, which can improve symptoms of irritable bowel syndrome and with enhanced nutritional value. In addition, it is well recognized that L. plantarum showed probiotic effect in modulation of gut microbiota. Since the literature of this important aspects is scarce, I suggest enriching the scientific relevance. Perhaps these articles are helpful to be cited and discussed:
- Caponio, G.R., Difonzo, G., de Gennaro, G., Calasso, M., De Angelis, M., & Pasqualone, A. (2022). Nutritional improvement of gluten-free breadsticks by olive cake addition and sourdough fermentation: How texture, sensory, and aromatic profile were affected?. Frontiers in Nutrition, 9, 830932.
- Coda, R., Cassone, A., Rizzello, C. G., Nionelli, L., Cardinali, G., & Gobbetti, M. (2011). Antifungal activity of Wickerhamomyces anomalus and Lactobacillus plantarum during sourdough fermentation: identification of novel compounds and long-term effect during storage of wheat bread. Applied and environmental microbiology, 77(10), 3484-3492.
- Lippolis, T., Cofano, M., Caponio, G. R., De Nunzio, V., & Notarnicola, M. (2023). Bioaccessibility and Bioavailability of Diet Polyphenols and Their Modulation of Gut Microbiota. International Journal of Molecular Sciences, 24(4), 3813
-Caponio, G. R., Noviello, M., Calabrese, F. M., Gambacorta, G., Giannelli, G., & De Angelis, M. (2022). Effects of grape pomace polyphenols and in vitro gastrointestinal digestion on antimicrobial activity: Recovery of bioactive compounds. Antioxidants, 11(3), 567.
3. Results and Discussion:
-lines 278, 283: please double check the technical quality of Figure 2 and 3 (some parts appear blurred).
-The paragraph 3.2. needs more discussion.
-lines 304 and 321: replace “asterisks denote..” with “* means statistically..”
Minor editing of English language required
Author Response
Dear Reviewer,
We have received your comments on our submitted original manuscript and appreciate the constructive and helpful recommendations. In accordance with the given comments, the manuscript has been improved. In addition to submitting a revised manuscript (a marked-up copy), we are providing a point-by-point response to your comments outlined below.
- Keywords: -Please write in alphabetic order.
R: In the “Instructions for Authors” there is no such a recommendation. In published articles keywords are written in the undirected order. In our paper keywords are written consistently with the results of the work.
- Introduction:
-lines 29: write the abbreviation L. plantarum in brackets for the first time, and then always use the abbreviated form. The same in abstract (line 10).
R: Names of microbes are written in the manuscript in accordance with the rules of nomenclature of bacteria. Namely, if the genus name has already been stated, it is further abbreviated to the first letter of the generic name. If species are listed belonging to several genera which have the same initial letter, the generic name is used in full (Cowan M. Kelly. Microbiology: a systems approach, 3rd ed., McGraw-Hill. 2012; Lapage SP, Sneath PHA, Lessel EF, et al., editors. International Code of Nomenclature of Bacteria: Bacteriological Code, 1990 Revision. Washington (DC): ASM Press; 1992, https://www.ncbi.nlm.nih.gov/books/NBK8816/). Corresponding corrections were done in lines 76, 313, and 319.
-lines 47-50: The health-promoting effects of L. plantarum are extensively studied in view of innovative valorisation strategies including the bioconversion of L. plantarum plant by-products to obtain enzymes, polysaccharides, nutraceuticals and beverages. It is also important in sourdough fermentation-based strategies for making baked goods, which can improve symptoms of irritable bowel syndrome and with enhanced nutritional value. In addition, it is well recognized that L. plantarum showed probiotic effect in modulation of gut microbiota. Since the literature of this important aspects is scarce, I suggest enriching the scientific relevance. Perhaps these articles are helpful to be cited and discussed:
- Caponio, G.R., Difonzo, G., de Gennaro, G., Calasso, M., De Angelis, M., & Pasqualone, A. (2022). Nutritional improvement of gluten-free breadsticks by olive cake addition and sourdough fermentation: How texture, sensory, and aromatic profile were affected?. Frontiers in Nutrition, 9, 830932.
- Coda, R., Cassone, A., Rizzello, C. G., Nionelli, L., Cardinali, G., & Gobbetti, M. (2011). Antifungal activity of Wickerhamomyces anomalus and Lactobacillus plantarum during sourdough fermentation: identification of novel compounds and long-term effect during storage of wheat bread. Applied and environmental microbiology, 77(10), 3484-3492.
- Lippolis, T., Cofano, M., Caponio, G. R., De Nunzio, V., & Notarnicola, M. (2023). Bioaccessibility and Bioavailability of Diet Polyphenols and Their Modulation of Gut Microbiota. International Journal of Molecular Sciences, 24(4), 3813
-Caponio, G. R., Noviello, M., Calabrese, F. M., Gambacorta, G., Giannelli, G., & De Angelis, M. (2022). Effects of grape pomace polyphenols and in vitro gastrointestinal digestion on antimicrobial activity: Recovery of bioactive compounds. Antioxidants, 11(3), 567.
R: Thank you. The comment is reasonable and we appreciate the opportunity to clarify our view. The health-promoting effects of L. plantarum are extensively studied and therefore in the Introduction we have to cite mainly review papers to cover as much information as possible. The ability of L. plantarum to improve symptoms of irritable bowel syndrome was indicated in lines 46-47. Modulation of gut microbiota by L. plantarum was considered as one of the mechanisms through which this species realizes its beneficial effect on the human body (line 50). Indeed, fermentation of different plant substrates by L. plantarum my favourably alter the properties of these substrates. The publications offered by the Reviewer substantively contribute to the unraveling of this aspect of biological activity of L. plantarum, but cannot be cited in this manuscript because are only remotely related to the main topic of our study. We implied the variety of applications of L. plantarum in food production in line 31 (“…widely used in food ..”) and lines 37-38 (“L. plantarum is valued in the food industry for its ability to alter favorably the texture, flavor, nutritional quality, and the shelf life of foods [4].”).
- Results and Discussion:
-lines 278, 283: please double check the technical quality of Figure 2 and 3 (some parts appear blurred).
R: Thank you. We have converted figures 2-4 into TIFF format and checked the resolution.
-The paragraph 3.2. needs more discussion.
R: Thank you. Discussion was added into paragraph 3.2.
-lines 304 and 321: replace “asterisks denote..” with “* means statistically..”
R: Thank you. It was corrected.